# Immunotherapy for Prostate Cancer: A Current Systematic Review and Patient Centric Perspectives

**DOI:** 10.3390/jcm12041446

**Published:** 2023-02-11

**Authors:** Laeeq ur Rehman, Muhammad Hassan Nisar, Wajeeha Fatima, Azza Sarfraz, Nishwa Azeem, Zouina Sarfraz, Karla Robles-Velasco, Ivan Cherrez-Ojeda

**Affiliations:** 1Department of Research, Continental Medical College, Lahore 54000, Pakistan; 2Department of Research, Aziz Fatima Medical and Dental College, Faisalabad 38000, Pakistan; 3Department of Research, Quaid e Azam Medical College, Bahawalpur 63100, Pakistan; 4Department of Pediatrics and Child Health, The Aga Khan University, Karachi 74800, Pakistan; 5Schwarzman College, Tsinghua University, Beijing 100190, China; 6Department of Research and Publications, Fatima Jinnah Medical University, Lahore 54000, Pakistan; 7Department of Allergy, Immunology and Pulmonary Medicine, Universidad Espíritu Santo, Samborondón 092301, Ecuador

**Keywords:** prostate cancer, immunotherapy, clinical trials, therapeutic developments, men’s health

## Abstract

Prostate cancer is the most commonly diagnosed cancer in men worldwide, making up 21% of all cancer cases. With 345,000 deaths per year owing to the disease, there is an urgent need to optimize prostate cancer care. This systematic review collated and synthesized findings of completed Phase III clinical trials administering immunotherapy; a current clinical trial index (2022) of all ongoing Phase I–III clinical trial records was also formulated. A total of four Phase III clinical trials with 3588 participants were included administering DCVAC, ipilimumab, personalized peptide vaccine, and the PROSTVAC vaccine. In this original research article, promising results were seen for ipilimumab intervention, with improved overall survival trends. A total of 68 ongoing trial records pooling in 7923 participants were included, spanning completion until June 2028. Immunotherapy is an emerging option for patients with prostate cancer, with immune checkpoint inhibitors and adjuvant therapies forming a large part of the emerging landscape. With various ongoing trials, the characteristics and premises of the prospective findings will be key in improving outcomes in the future.

## 1. Introduction

### 1.1. Brief Overview

Prostate cancer (PC) is the most commonly diagnosed cancer in men and is the second most commonly occurring disease among males in the United States (US) [1]. In 2022 alone, 268,490 new cases of PC occurred in the US [1]. PC makes up around 21% of all cancer cases in males [1]. The disease leads to 345,000 deaths per year; it is the second most common cancer-causing death in the US following lung and bronchus cancer. PC is often termed a ‘cold tumor’ given its immunosuppressive microenvironment [2]. The tumor-infiltrating lymphocytes inhibit T effecter cell activity, thereby contributing to the progression of PC. Biopsy specimens have depicted that the infiltrating lymphocytes skew towards T helper 17 and T regulatory phenotypes that suppress the body’s antitumor immune and autoreactive T cell responses [2,3]. There is a growing need to design therapies that can boost immunity with effector T cells and antigen-presenting cells [4]. The subgroup of antigen-presenting cells, dendritic cells, are notable CD8+ T cells that can be used in activating and killing tumors [5]. Studies show associations of positive prognosis with dendritic cell tumor infiltration [6]. Androgen deprivation therapy (ADT) has also led to the mitigation of T cell tolerance along with the priming of T cells to prostatic antigens. These developments are suggestive of the synergistic combination of immunotherapy with ADT.

With immunotherapy and precision medicine penetrating oncological care, novel immunotherapeutic approaches have become a part of standard care in recent years [7]. The treatment modality has shown promising outcomes among patients with aggressive cancers, with long-term remission becoming a possibility [8]. Prostate cancer advancements include sipuleucel-T and immune checkpoint inhibitors (ICIs), which provide alternatives for castration-resistant PC coupled with chemotherapy and ADT [9,10,11]. Immunotherapy intends to target cancer cells by recognizing T cells or antibodies [12]. However, given the immunosuppressive state of prostate cancer cells, the immune responses to treatment have been less effective when compared to melanoma, renal cell carcinoma, head and neck cancer, and non-small-cell lung cancer [13,14,15,16,17,18]. To overcome the suppressive tumor microenvironment (TME), immunotherapy trials aim to target the infiltration of T cells, the mutational burden of prostate cancer cells, and the combined efficacy of treatment [19]. Recently, among the special subgroup of patients that present with high PD-L1 tumor expression, CDK12 mutations, or high microsatellite instability (MSI) and mismatch repair deficiency (dMMR), ICIs may be key in inciting responses to combination therapy [20,21,22,23].

### 1.2. Rationale

Immunotherapy remains to be a momentous area of prostate cancer care, and is an appealing treatment paradigm in optimizing the management of the disease. Despite obtaining success against other cancer types, prostate cancer has so far shown mixed findings with immunotherapy. With the first-ever prostate cancer vaccine approved in 2010, patients with advanced prostate cancer were provided with a viable treatment modality to improve outcomes of disease. However, the spileucel-T vaccine has only partially improved survival outcomes. Thereby, the purpose of this study is to provide readers with an updated view of trials specifically in Phase III of testing, since these trials test if the novel immunotherapy is better than standard treatment. On the other hand, Phase I trials test only the safety of new therapies, while Phase II trials tend to assess efficacy of the new treatment among patients with prostate cancer. This systematic review will include current literature comprising Phase III trials that are key in navigating the direction of patient care. At present, there are only three FDA-approved immunotherapy options for adult male patients with prostate cancer. These include sipuleucel-T, which is a vaccine made with patients’ immune cells that have been stimulated to target the prostatic acid phosphatase (PAP) protein. This is approved for only the subset of patients with advanced prostate cancer. The other two options comprise immunomodulating therapies, including dostarlimab and pembrolizumab. Both of these are immune checkpoint inhibitors that target the PD-1/PD-L1 pathways; these are approved among the subset of patients with DNA mismatch repair deficiency (dMMR), microsatellite instability (MSI-H), or high mutational burden (TMB-H). Notably, the FDA has approved six drugs since 2017 which have histology-agnostic indications of interest in metastatic castration-resistant PC [24]. These include pembrolizumab (tumors with dMMR/high MSI), dostarlimab (dMMR tumors), entrectinib and lartotrectinib (tumors with neurotrophic tyrosine receptor kinase fusions), and trametinib combined with dabrafenib (tumors with BRAF V600E mutations) [24].

### 1.3. Aims and Objectives

While three immunotherapies are approved for prostate cancer and are being administered among patients that fulfil the criteria of administration, the aims and objectives of this systematic review are to collate evidence for patients with any stage/grade of prostate cancer, being intervened either with immunotherapy alone or in combination compared with control/standard care. There are three primary outcomes of interest; these include progression-free survival (PFS), overall survival (OS), and response rate (RR). We will firstly collate and synthesize findings of all completed Phase III clinical trials administering immunotherapy to patients with prostate cancer. Secondly, we will present a current clinical trial index (2022) of all Phase I–III clinical trial records that are ongoing in the field.

## 2. Methods

### 2.1. Literature Search

To obtain completed Phase III clinical trials, a systematic search was conducted in PubMed/MEDLINE, Embase, Scopus, and CINAHL adhering to PRISMA Statement 2020 guidelines. The search was conducted from inception until 20 November 2022. An additional search was conducted in Elsevier, *BMJ*, *JAMA*, *NEJM*, and *The Lancet* to locate relevant literature; this methodology is referred to as handsearching and is utilized to identify any additional randomized, controlled trials administering immunotherapy to patients with prostate cancer. To identify ongoing prostate cancer immunotherapy clinical trials in Phase I–III, a systematic search was conducted in ClinicalTrials.Gov and the World Health Organization’s International Clinical Trial Registry Platform (ICTRP); both engines were searched until 20 November 2022. A combination of the following keywords was applied across the databases and search engines: immunotherapy, prostate, cancer, neoplasm, carcinoma, clinical, and trial. The search string is attached in the Appendix A. Gray literature was not included in this study. The PICO framework for this systematic review is as follows:Participants: Adult patients with prostate cancer;Intervention: Any form of immunotherapy;Comparator: Standard care (chemotherapy, radiotherapy, surgery) or placebo;Outcome: Any form of survival, progression, responder rate, adverse events, or other treatment outcomes.

### 2.2. Eligibility Criteria

This study is divided into two parts. The first is a systematic assessment of Phase III completed clinical trials. The second is Phase I–III ongoing clinical trial records, presented systematically as an index for readers.

Clinical trials were the only study and record type that were considered for this study. No language restrictions were placed. All non-English-language studies were translated into English using Google Translate. Cohorts, case controls, case reports, brief reports, systematic reviews, and meta-analytical studies were omitted.

The participants were male adults, with prostate cancer at a local, metastatic, or any stage of progression, being intervened with immunotherapy alone or in combination with standard-of-care therapies, with outcomes of survival, progression-free disease, adverse events, or other key indicators or prognosis of treatment.

### 2.3. Study Selection

The title and abstract screening in addition to the full-text screening was led by two mid-career researchers (Z.S. and A.S.) independently. Any disagreements were resolved through discussion with a third researcher (I.C.-O). The data extraction was performed by all researchers and was rechecked independently by Z.S. in the shared spreadsheets, which were first tested and adapted on sample studies. The studies’ bibliographic data was stored in EndNote X9 (Clarivate Analytics). The reference management software employed in this study was Mendeley (Elsevier, Amsterdam, The Netherlands).

### 2.4. Data Extraction

The data for completed clinical trials were extracted as number, author and year, title, journal, phase, design, inclusion criteria, intervention, primary outcome measures, follow-up, sample size, efficacy outcomes, and remarks.

For ongoing trials, the data were extracted in two parts. The first part comprised NCT number, status, conditions, interventions, and outcome measures. The second part consisted of NCT number, phase, age, enrollment, study type, study design, completion date, collaborators, and location.

### 2.5. Risk of Bias Assessment

The bias among the completed clinical trials was assessed using Version 2 of the Cochrane risk-of-bias tool for randomized trials (RoB 2). The RoB 2.0 assessment comprises the following five domains of bias: randomization process, deviations from intended interventions, missing outcome data, measurement of the outcome, and selection of the reported result. Domain-level judgments about risk of bias were classified as low risk of bias, some concerns, and high risk of bias. The traffic light plot of bias assessment and the weighted summary plot of the overall type of bias are illustrated in Section 3.3: risk of bias synthesis.

### 2.6. Protocol Registration and Role of Funding

The protocol of this systematic review was registered with Open Science Framework (OSF): osf.io/4vs7w. No funding was obtained.

## 3. Results

During the identification of studies via databases, a total of 3808 studies were identified, of which 467 duplicates were removed. A total of 3341 studies were screened with titles and abstracts, after which 3135 met the exclusion criteria. Finally, 206 full-text studies were assessed, of which four Phase III clinical trials were included in this systematic review. During the identification phase of clinical trial records, a total of 318 were identified from websites. Of these, 208 records were sought for retrieval and assessed for eligibility. Of them, 68 records were included in this systematic review. The PRISMA flowchart depicting the study selection process is illustrated in Figure 1.

### 3.1. Phase III Clinical Trials

Four Phase III trials of immunotherapy for prostate cancer were included [25,26,27,28]. A total of 3588 participants were enrolled across all trials. The designs were randomized and controlled with standard-of-care approaches in all of the included studies. The choice of interventions comprised autologous dendritic cell-based immunotherapy (DCVAC), intravenous ipilimumab therapy, personalized peptide vaccination, and PROSTVAC (a vaccine). Individual trial findings are further described below and are tabulated in Table 1.

Vogelzang and colleagues identified the efficacy and safety of autologous dendritic cell-based immunotherapy (DCVAC) among metastatic castration-resistant prostate cancer with a castration period of over 4 months [25]. DCVAC was an add-on and maintenance given every 3–4 weeks for up to 15 doses. The primary outcome measure was overall survival. The Phase III double-blind, parallel-group, placebo-controlled, randomized trial enrolled 1182 participants with a follow-up period of 58 months. The trial did not meet its outcome measures given that there were no differences in median OS between DCVAC and placebo groups reported at 23.9 months and 24.3 months, respectively. With an HR of 1.04, there was no notable difference in the likelihood of death in either group.

Fizazi et al. conducted a final analysis of their Phase III trial, which administered ipilimumab intravenously post bone-directed radiotherapy or among non-progressing prostate cancer patients [26]. With a total of 799 patients enrolled, the patients were followed for 2.4 years, with a single primary outcome of overall survival. The overall survival rates were higher in the ipilimumab group compared to placebo at 2 (25.2% and 16.6%) and 5 years (1.9% and 2.7%), respectively. One caveat is that 1.8% of patients in the ipilimumab group and 0.3% in the placebo group died due to study drug toxicity.

Noguchi et al. (2021) conducted a randomized, double-blind, placebo-controlled trial of personalized peptide vaccination for castration-resistant prostate cancer after receiving docetaxel [27]. The 310 patients enrolled in the trial were HLA-A24 positive and were stratified as aged less than and more than 75 years. The primary outcome measure of increasing overall survival in the vaccine group was not met, with 16.1 months with intervention and 16.9 months in the standard care group.

Gulley et al. (2019) conducted a Phase III trial of PROSTVAC, a vaccine designed to enable the immune system to recognize and attack prostate cancer cells [28]. PROSTVAC (250 ug, lyophilized) was combined with GM-CSF in one arm; PROSTVAC was given alone in the second arm; the third arm received a placebo. The primary outcome measure was overall survival, with follow-ups made until 25 weeks. None of the active treatment arms yielded an effect on median overall survival rates, with 34.4 months for the first arm, 33.2 months for the second arm, and 34.3 months for the placebo.

### 3.2. Ongoing Clinical Trials

We located a total of 19 ongoing Phase I clinical trials. The total enrollment was 1989 individuals. The trials had a completion date spanning December 2022 until December 2027. The locations included Belgium, China, France, Italy, Korea, Spain, the United States, and the United Kingdom. The conditions included castration-resistant prostate cancer (CRPC), metastatic castrate-resistant prostate cancer (mCRPC), metastatic castration-resistant prostate adenocarcinoma, aggressive variant PC, stage III/IIIA/IIIB/IIIC/IV/IVA/IVB PC AJCC v8, variants with testosterone greater than 150 ng/dL, and recurrent prostate cancer. The interventions included AZD4635, Abiraterone Acetate, Acapatamab, ADXS-504, AMG 509, BMS-986218, Cabozantinib S-malate, CAR-T cell immunotherapy, CCW702, CDX-301, Cellgram-DC-PC, CT-0508, Cytochrome P450 (CYP) Cocktail, Daratumumab, Degarelix, Durvalumab, engineered autologous T cells, Enzalutamide, Etanercept, FMS inhibitor JNJ-40346527, INCB106385, INCMGA00012, Ipilimumab, Nivolumab, Oleclumab, Pegilodecakin, Pembrolizumab, PGV-001, Poly-ICLC, Valemetostat, and VMD-928. Procedures included radical prostatectomy, peripheral blood/biospecimen collection, and magnetic resonance imaging. The characteristics of current clinical trials at Phase I are enlisted in Table 2. The enrollment, study design, completion date, collaborators, and key locations of trials at Phase I are depicted in Table 3.

A total of 14 ongoing trials were located at Phase I/II for prostate cancer. The total enrollment was 1079 participants. The trials were conducted in Australia, France, Germany, Hungary, the United Kingdom, and the United States. The trials spanned completion between December 2022 and June 2028. The conditions included mCRPC, CRPC, metastatic malignant neoplasms in the lymph nodes, Stage IV/IVA/IVB PC AJCC v8, and prostate carcinoma metastatic in the bone. The interventions comprised 177Lu-PSMA, 225Ac-J591, androgen deprivation therapy (ADT), Atezolizumab, Avelumab, BMS-986253, BNT112, Cemiplimab, Degarelix, Durvalumab, Epacadostat, FPV-Brachyury, HB-302/HB-301 Alternating 2-Vector Therapy, Ipilimumab, Ivuxolimab, M7824, Metronomic Vinorelbine, MVA-BN-Brachyury, N-803, Nivolumab Pembrolizumab, Peposertib, PROSTVAC-V/F, Radiation Therapy (Radium Ra 223 Dichloride, Brachytherapy, External Beam Radiation Therapy, Tivozanib, Tremelimumab, and Utomilumab. Quality-of-life assessments and diagnostic testing (i.e., 68Ga-PSMA-11) were also conducted. The characteristics of current clinical trials at Phase I/II are enlisted in Table 4. The enrollment, study design, completion date, collaborators, and key locations of trials at Phase I/II are depicted in Table 5.

A total of 35 ongoing clinical trials were located in Phases II and III. The completion dates spanned December 2022 to January 2028. A total of 4855 participants were enrolled. The trials were conducted in Argentina, Australia, Austria, Belgium, Canada, Czechia, France, Germany, Italy, Japan, Mexico, Netherlands, Puerto Rico, Singapore, Spain, Switzerland, Taiwan, and the United States. The conditions included mCRPC, advanced prostate/metastatic cancer, localized PC, castration-sensitive PC, prostate adenocarcinoma, prostatic neoplasms, locally advanced PC, and Stage IV PC AJCC v8. The interventions comprised the following: Prednisone, Abemaciclib, Abiraterone Acetate, Adavosertib, Aglatimagene besadenovec, androgen deprivation therapy (ADT), Apalutamide, Atezolizumab, Bintrafusp alfa, BN-Brachyury, Cabozantinib S-malate, Cetrelimab, CFI-400945, CV301, Darolutamide, Degarelix, Durvalumab, Enzalutamide, Etrumadenant, Ipatasertib, Ipilimumab, M9241, MSB0011359C (M7824), N-803, NIR178, Niraparib, Nivolumab, Olaparib, PDR001, Pembrolizumab, PROSTVAC-F, PROSTVAC-V, pTVG-AR, pTVG-HP, Radiation (Stereotactic Body Radiation Therapy-SBRT), Savolitinib, Sipuleucel-T, SRF617, SV-101, SV-102, Tremelimumab, and Zimberelimab. The characteristics of current clinical trials at Phases II and III are enlisted in Table 6. The enrollment, study design, completion date, collaborators, and key locations of trials at Phases II and III are depicted in Table 7.

### 3.3. Risk-of-Bias Synthesis

On noting the bias arising from the randomization process, all four RCTs had low concerns. The risk of bias due to deviation from the intended intervention was low in all of the included studies. On assessing bias due to missing outcome data, two RCTs had some concerns, whereas two had low concerns. When noting bias in the measurement of the outcome, all RCTs had low concerns. For bias in the selection of the reported result, three RCTs had low concerns whereas one study had some concerns. Overall, three RCTs had low concerns for risk of bias while one RCT had some concerns (Figure 2).

## 4. Discussion

In this systematic review, a total of four Phase III trials administering immunotherapy to patients with prostate cancer were included. A total of 3588 participants were polled across these trials being administered DCVAC, ipilimumab, personalized peptide vaccine, and the PROSTVAC vaccine. Thus far, promising results of overall survival were seen with ipilimumab therapy (25.2% overall survival in the intervention group compared to 16.6% in placebo) [26]. A total of 68 ongoing trials were tabulated and thereby discussed. These trials were currently pooling 7923 participants worldwide, spanning completion until June 2028.

The past decade has led to the development of immune checkpoint inhibitors (ICIs) for prostate cancer [29,30]. While numerous Phase III clinical trials have provided mixed prognostic findings, ICIs–including pembrolizumab, approved by the FDA in 2017–have been utilized in clinical trials, but have only prevented DNA repair in less than 5% of men with advanced prostate cancer [31,32,33]. Therapeutic cancer vaccines, including sipuleucel-T, PROSTVAC, and personalized peptide vaccines, have not led to significant survival differences in patient populations [34,35,36]. Newer trials combining vaccines and other agents, the immune response, and ICIs may be able to downgrade the tumor defenses against T cells [37,38,39,40]. Sipuleucel-T did, however, lead to differences in T cells that were thrice activated in vaccinated patients as compared to placebo groups; therefore, the vaccine may prime patients’ immune response [41,42,43,44]. The PROSTVAC (PSA-TRICOM) vaccine was another variant utilizing the poxvirus to deliver genes to spur molecular production of T cells and improve the targeting of PSA [28,45,46,47]. However, the Phase III trial’s findings in 2019 were unfavorable in infiltrating the tumor, despite generating an immune response [28]. PROSTVAC is currently being tested in men with locally advanced prostate cancer along with PD1 inhibitors [48,49,50,51,52]. A small cohort of a clinical trial in progress has revealed that two out of six participants showed PSA level reductions by more than 90% and one of six participants showed no evidence of disease during the 5 years [53]. The evidence suggests that combination immunotherapy increased CD4+ T cell density in the invasive margin with similar trends noted in the intratumoral and benign compartments [53]. The CD8+ T cell density also increased in the benign and invasive margins. T regulatory cells were present in low frequencies in the tumor immune microenvironment, and the Ki67 tumor cells dropped after treatment, suggesting that combination may control tumor growth [53]. The neoadjuvant PROSTVAC and nivolumab may lead to increased infiltration of immune cells [54,55,56]. The combination is being tested to control prostate cancer growth [53].

Other combinations of vaccines including the mRNA variant are being tested with ICIs, including cemiplimab, which is currently approved for skin cancer [57,58,59,60]. mRNA vaccines are also being combined with androgen receptors and with pembrolizumab [61,62,63]. Other trials have combined PROSTVAC with ipilimumab, a monoclonal antibody that targets CTLA-4, which is a protein located on regulatory T cells and can deactivate other T cells [45,47,61,64]. Experimental testing has also steered efforts in adding a third modality of a cytokine, interleukin-15, to target immune signaling to target natural killer cells [65,66,67,68]. The QuEST1 study showed that the triple-hit approach of BNVax (a therapeutic poxviral vaccine targeting brachyury), anti-PD-L1 monoclonal antibodies, and interleukin-15 superagonist complexes have eradicated traces of bone detectability of bony metastasis in two patients with metastatic disease [69]. The tripartite therapy is experimental and the QuEST1 study interrogated the safety and efficacy of immunotherapy combinations for CRPC [69].

The combination of ICIs and vaccines is not the only modality of current immunotherapy paradigms. CAR-T cell therapy is also being deployed in the early clinical trial setting; it comprises T cell extraction from the patient and engineering to target specific cancer cells and reverse administering them to the individual [70,71,72,73]. The modality has been successful in cancers of hematological origin [74,75,76]. CAR-T cell therapy is being tested in prostate cancer research centers [77,78,79,80]; a recent report identifies 13 patients being treated with engineered CAR-T cells to target prostate-specific membrane antigens (PSMA) [78]. While PSMA is rarely found in many tissues, it is located near 80% of prostate cancer cells and increases in prevalence as cancer progresses. Three of the 13 participants had a 30% reduction in PSA levels; however, five patients experienced cytokine release syndrome, which is an inflammatory reaction to treatment; one patient died [78]. Another trial was halted due to the neurotoxic side effects of CAR-T cell therapy [81,82,83]. This has led to the consideration of selectively injecting CAR-T cells into the tumor directly as compared to system administration, which has led to mostly adverse outcomes [84,85].

Another treatment modality is the bi-specific T cell engagement (BiTE), which are monoclonal antibodies with two hooks [86,87,88]. One hook is for the protein outside the tumor cells whereas the second hook is for the T cell surface receptor, CD3; BiTE brings the two cells together. BiTE is currently under investigation with acapatamab (AMG 160), with response rates to Phase I trials approaching 33% [89]. The modality is being combined with PD1 blockers and hormone therapy. However, a common adverse event is cytokine storm syndrome, which is the double-edged sword of immunotherapeutic treatment [90,91,92]. Newer formulations of molecules with lower affinity for CD3 may help in overcoming the cytokine storm among patients [93,94].

For patients to receive beneficial immunotherapies, the patient groups must be segregated based on the immunogenicity of individual diseases. The consensus is that prostate cancer may respond to immunotherapy approaches once the patient populations are personalized; this has been noticed in skin, kidney, and breast cancers, but has not been the present reality of prostate cancer. Immunotherapy is also believed to only work among a small group of men whose tumors fit narrow inclusion criteria based on molecular and pathological factors. However, once an effective combination is tested in combination regimens, the therapy can reach a larger scale, insofar as adverse events, including neurotoxicity and cytokine storm-like responses, have hindered scalability. Another caveat is that immunological interventions have largely been administered to patients with advanced disease only; however, with the progression of the disease, the T cells decrease in count. It may be worthwhile to deploy immunotherapy at an earlier stage of the disease or immediately after radiation or surgical interventions. Radiotherapy may also act as a primer of the immune system, thereby allowing immune responses to be more effective. The consideration of administering immunotherapy before ADT is also existent. Immunotherapy may lower testosterone levels, allow T cells to circulate in the prostate gland, release inflammatory cytokines, and reduce the need for hormone therapy altogether. The decision can be reviewed if immunotherapy does not work; the first choice of hormone therapy typically leads to fatigue, weight gain, and muscle loss. One Phase II trial of pembrolizumab and enzalutamide (androgen-receptor blocker) presented exceptional responses in five out of 20 participants, despite body metastasis present in two of the responders to treatment [95]. Therefore, immunotherapy approaches must ideally target bone tumors as well.

### 4.1. Limitations

Our study has certain limitations that future studies must address. Firstly, given the nature of this study, the number of completed clinical trials is relatively small. Secondly, our criteria were to only include clinically relevant Phase III trials to make them useful for the patient population. To address this, we included all ongoing trials being conducted in the arena of prostate cancer and immunotherapy. Lastly, we utilized Google Translate during the study selection process to screen and include studies; this was in lieu of using interpreters specialized in medical research.

### 4.2. Future Directions

Emerging evidence points towards cytokines and chemokines as key players of the pleiotropic actions of PC—such as angiogenesis, growth, endothelial mesenchymal transition, leukocyte infiltration, and hormone escape for advanced cases. As a result, the chemokine system and immune cells are key targets to be scaled in suppressing tumorigenic environments while serving as potential immunotherapy for prostate cancer [96]. There has been sanguinity towards prostate cancer immunotherapy based on small-scale clinical trials. The recent development of CAR-T therapy has also revolutionized the treatment of resistant malignancies, with many studies underway utilizing this technology in treating solid tumors [97]. It is yet to be determined if immunotherapies either alone or in combination can lead to remission in patients with advanced prostate cancer. There is cautious optimism about the path ahead.

## 5. Conclusions

Completed trials using immunotherapy with vaccines and immune checkpoint inhibitors have so far been unable to make a breakthrough in the treatment of patients with advanced prostate cancer. Proof-of-concept studies, however, have shown success among select responders by inducing immunologic responses. Immunotherapy is an emerging option for treating patients with prostate cancer. Various obstacles have been noted with current immunotherapies, including mRNA vaccines, CAR-T cell therapy, and PD-1 blockers. Overall, ICIs, and neo- and adjuvant therapies form a large part of the emerging landscape. The timing of commencing immunotherapy has also led to baffling findings. With 68 ongoing trials of immunotherapy and prostate cancer, the characteristics and premises of the prospective findings will be key in improving outcomes in the near future.

## Figures and Tables

**Figure 1 jcm-12-01446-f001:**
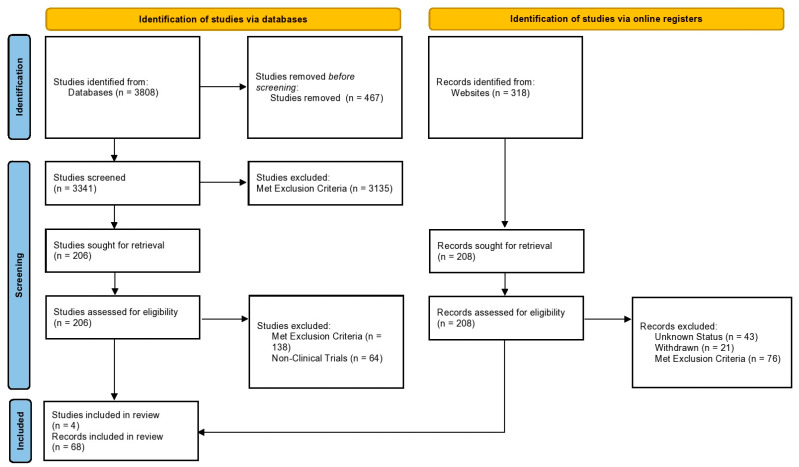
PRISMA flowchart depicting the study selection process.

**Figure 2 jcm-12-01446-f002:**
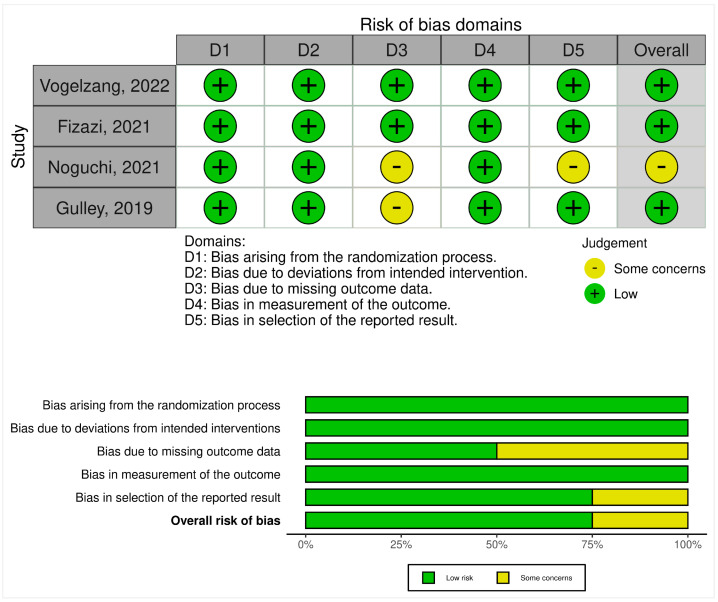
Risk-of-bias assessment of RCTs using the ROB-2 tool. Traffic light plot of study-by-study bias assessment. Weighted summary plot of the overall type of bias encountered in all studies [25,26,27,28].

**Table 1 jcm-12-01446-t001:** Characteristics and Efficacy Outcomes of Completed Phase III Prostate Cancer-Immunotherapy Trials.

No	Author, Year	Journal	Phase	Design	Inclusion Criteria	Intervention	1° Outcome Measure(s)	Follow-Up	Sample Size	Efficacy Outcomes	Remarks
1	Vogelzang, 2022 [25]	*JAMA Oncology*	Phase 3	Double-blind, parallel-group, placebo-controlled, RCT; NCT02111577	mCRPC with >4 months of castration period	DCVAC/PCa (add-on and maintenance) every 3–4 weeks up to 15 doses or placebo; in combination with chemotherapy (docetaxel plus prednisone)	OS	58 months	1182	No difference in median OS between DCVAC/PCa (23.9 months, 95% CI = 21.6–25.3) and placebo groups (24.3 months, 95% CI = 22.6–26); HR = 1.04; 95% CI, 0.90–1.21; *p* = 0.6	Primary objectives were unmet; treatment-emergent adverse events in DCVAC/PCa (9.2%) or placebo (12.7%) were comparable
2	Fizazi, 2021 [26]	*European Urology*	Phase 3	Double-blind RCT (CA184–043); NCT00861614	mCRPC, histologically/cytologically confirmed adenocarcinoma; ≥1 bone metastases, testosterone < 0.50 ng/mL	Bone-directed RT followed by ipilimumab IV (10 mg/kg) or placebo every 3 weeks (up to four doses); non-progressing patients: ipilimumab (10 mg/kg) or placebo as maintenance therapy every 3 months	OS	2.4 years	799	OS rates were higher in the ipilimumab vs. placebo arms at 2 years: 25.2% vs. 16.6%, 3 years: 15.3% vs. 7.9%, 4 years: 10.1% vs. 3.3%, and 5 years: 7.9% vs. 2.7%	Disease progression was the most frequent cause of death in both arms; seven patients (1.8%) in the ipilimumab arm and one patient (0.3%) in the placebo arm died due to study drug toxicity
3	Noguchi, 2021 [27]	*Oncology Reports*	Phase 3	Double-blind RCT; UMIN000011308	HLA-A24-positive patients with castration-resistant PC; within 12 months of docetaxel chemotherapy	Personalized peptide vaccination or placebo using the minimization technique with age stratification factors: <75 or ≥75, and use of enzalutamide or abiraterone (with or without)	OS	29.8 months and 27.4 months	310	The estimated median OS was 16.1 months (95% CI= 13–18.2) with PPV and 16.9 months (95% CI = 13.1–20.4) with placebo; HR = 1.04 (95% CI = 0.8–1.37, *p* = 0.77)	Both groups experience the same proportion of grade ≥ three adverse events (41%); OS did not expand with peptide vaccine therapy in HLA-A24-positive patients
4	Gulley, 2019 [28]	*Journal of Clinical Oncology*	Phase 3	Multicenter, double-blind RCT; NCT01322490	Progressive mCRPC, testosterone level < 50 ng/dL, current use of a GnRH agonist/antagonist (unless surgically castrated), chemotherapy naïve for metastatic PC	PROSTVAC plus GM-CSF, 250 μg, lyophilized (Arm VG), PROSTVAC plus placebo GM-CSF (Arm V), or vaccine placebo plus placebo GM-CSF (Arm P)	OS	25 weeks	1297	None of the active treatment arms had an effect on median OS; Arm V, OS = 34.4 months, HR = 1.01 (95% CI = 0.84–1.2, *p* = 0.47); Arm VG, OS = 33.2 months, HR = 1.02 (95% CI = 0.86–1.22, *p* = 0.59); Placebo = 34.3 months	While PROSTVAC was well tolerated with <1% adverse events, no benefits were present for OS

Abbreviations: 68Ga: gallium-68; ADT: androgen deprivation therapy; CI: confidence interval; DCVAC: autologous dendritic cell immunotherapy; ELISPOT: enzyme-linked immunospot; FLT: 3′-Deoxy-3′-^18^F-fluorothymidine; GnRH: gonadotropin-releasing hormone; HR: hazard ratio; IRAEs: immune response adverse events; mCRPC: metastatic castration-resistant prostate cancer; OS: overall survival; PAP: prostatic acid phosphatase; PAP-GM-CSF: prostatic acid phosphatase granulocyte-macrophage colony-stimulating factor; PCa: prostate cancer; PET: positron emission tomography; PFS: progression-free survival; PMSA: prostate-specific membrane antigen; PSA: prostate-specific antigen; RCT: randomized, controlled trial; rPFS: radiographic progression-free survival; RT: radiotherapy.

**Table 2 jcm-12-01446-t002:** Characteristics of Current Clinical Trials at Phase I for Prostate Cancer, 2022 (as of 5 December 2022).

No.	NCT Number	Status	Conditions	Interventions	Key Outcome Measures
1	NCT04301414	Recruiting	Prostate Cancer	BMS-986218 and Degarelix	Adverse events; pCR; PSA
2	NCT04615845	Recruiting	CRPC	Cellgram-DC-PC	Safety; immune response and tumor markers; PSA
3	NCT02740985	Active, not recruiting	mCRPC and others	AZD4635; Durvalumab; Abiraterone Acetate; Enzalutamide; Oleclumab; Docetaxel	DLTs; adverse events; pharmacokinetics; tumor Response; PFS
4	NCT04388852	Recruiting	CRPC; Metastatic Prostate Carcinoma; Stage IV, IVA, IVB Prostate Cancer AJCC v8	Ipilimumab; Valemetostat	Adverse events; immunologic and molecular effects; TTF; ORR
5	NCT04660929	Recruiting	HER2-positive; Prostate Cancer and various others	CT-0508	Safety and tolerability; ORR, PFS
6	NCT01140373	Active, not recruiting	Prostate Cancer	Engineered autologous T cells; Cyclophosphamide	Safety and tolerability; bone metastases/biomarkers of bone metastasis; humoral and cell-mediated immunity to PSMA and other known prostate cancer antigens; PSA; anti-PSMA autologous T cells
7	NCT03177460	Active, not recruiting	Prostate Adenocarcinoma; Stage III, IIIA, IIIB, IIIC Prostate Cancer AJCC v8; Testosterone Greater Than 150 ng/dL	Daratumumab; FMS Inhibitor JNJ-40346527; Radical Prostatectomy	Adverse events; pCR; immune changes in blood/tumor tissue
8	NCT02009449	Active, not recruiting	Prostate Cancer and various others	Pegilodecakin; Paclitaxel or Docetaxel and Carboplatin/Cisplatin; Oxaliplatin/Leucovorin/5-Fluorouracil; Gemcitabine/nab-paclitaxel; Capecitabine; Pazopanib; Pembrolizumab; Paclitaxel; Nivolumab; Gemcitabine/carboplatin	Safety and tolerability; adverse events; pharmacokinetics; change in tumor burden measured by volumetric CT/MRI; progression in bone-by-bone scintigraphy; anti-Pegilodecakin antibody formation
9	NCT04077021	Active, not recruiting	mCRPC, Adenocarcinoma	CCW702	Safety and tolerability; clinical efficacy
10	NCT04580485	Recruiting	CRPC and various others	INCB106385; INCMGA00012	Treatment-emergent adverse events; pharmacokinetic measures; ORR; DOR; Change in tumoral gene expression/immune cell activation
11	NCT03556228	Recruiting	Any Solid Tumors; Prostate Cancer and various others	VMD-928 300 and 100 mg tablet	Adverse events; pharmacokinetics; analgesic; change in TrkA protein expression; correlation between clinical antitumor/analgesic response and TrkA protein expression/AUC
12	NCT03805594	Active, not recruiting	CRPC; Metastatic PC; Prostate Adenocarcinoma; Stage IV, IVA, IVB Prostate Cancer	Lutetium Lu 177-PSMA-617; Pembrolizumab	ORR; adverse events; median DOR; PSA response rate; radiographic PFS; OS
13	NCT05077098	Recruiting	Recurrent Prostate Cancer	ADXS-504	Safety and tolerability; rates of treatment-related adverse events
14	NCT03792841	Active, not recruiting	mCRPC	Acapatamab; Pembrolizumab; Etanercept; Cytochrome P450 Cocktail	Treatment-emergent adverse events; pharmacokinetics; ORR; PSA; DOR; PFS; OS; CTC response
15	NCT04477512	Recruiting	Metastatic Hormone Refractory Prostate Cancer	Cabozantinib; Nivolumab; Abiraterone acetate; Prednisone	DLTs; PSA; ORR; OS; PFS; DSS; adverse events
16	NCT04514484	Recruiting	Advanced/recurred PC; CRPC-Metastatic PC; Stage IV Prostate Cancer AJCC v8, and various others	Cabozantinib S-malate; Nivolumab	DLTs; immune status
17	NCT05354375	Recruiting	Prostate Cancer	CAR-T cell immunotherapy	Safety; adverse events; PFS; OS
18	NCT04221542	Recruiting	Prostate Cancer	AMG 509; Abiraterone; Enzalutamide; Pembrolizumab	Adverse events; DLTs; pharmacokinetics; ORR; DOR; PSA; PFS; OS
19	NCT05010200	Recruiting	Prostate Cancer	PGV-001; Poly-ICLC; CDX-301	Adverse events; immune cell changes; radiographic-free survival

Abbreviations: CRPC: castration-resistant prostate cancer; CT: computed tomography; CTC: circulating tumor cells; DLTs: dose-limiting toxicities; DOR: duration of response; DSS: disease-specific survival; mCRPC: metastatic castrate-resistant prostate carcinoma; MRI: magnetic resonance imaging; ORR: overall response rate; OS: overall survival; pCR: pathological complete responses; PFS: progression-free survival; PSA: prostate-specific antigen; TTF: time to treatment failure.

**Table 3 jcm-12-01446-t003:** Enrollment, Study Design, Completion Date, Collaborators, and Key Locations of Current Clinical Trials at Phase I for Prostate Cancer, 2022 (as of 5 December 2022).

No.	NCT Number	Phases	Age	N	Study Type and Design	Completion Date	Collaborators	Locations
1	NCT04301414	Early Phase 1	≥18 years	24	Interventional; Randomized, Open Label, Treatment	May-24	Matthew Dallos; Bristol-Myers Squibb; Ferring Pharmaceuticals; Columbia University	United States
2	NCT04615845	Phase 1	20–80 years	10	Interventional; Single Group, Open Label, Treatment	Dec-22	Pharmicell Co., Ltd.	Republic of Korea
3	NCT02740985	Phase 1	18–130 years	313	Interventional; Non-Randomized, Open Label, Treatment	28-Dec-22	AstraZeneca	Various, United States
4	NCT04388852	Phase 1	≥18 years	80	Interventional; Single Group, Open Label, Treatment	31-Jan-23	M.D. Anderson Cancer Center; National Cancer Institute (NCI)	United States
5	NCT04660929	Phase 1	≥18 years	18	Interventional; Single Group, Open Label, Treatment	Feb-23	Carisma Therapeutics Inc	Various, United States
6	NCT01140373	Phase 1	≥18 years	13	Interventional; Single Group, Open Label, Treatment	Jun-23	Memorial Sloan Kettering Cancer Center; United States Department of Defense	United States
7	NCT03177460	Phase 1	≥18 years	33	Interventional; Non-Randomized, Open Label, Treatment	29-Jun-23	M.D. Anderson Cancer Center; National Cancer Institute (NCI)	United States
8	NCT02009449	Phase 1	≥18 years	350	Interventional; Non-Randomized, Single Group, Open Label, Treatment	17-Nov-23	Eli Lilly and Company; ARMO BioSciences	Various, United States
9	NCT04077021	Phase 1	≥18 years	22	Interventional; Non-Randomized, Open Label, Treatment	Dec-23	Calibr, a division of Scripps Research	United States
10	NCT04580485	Phase 1	≥18 years	230	Interventional; Non-Randomized, Parallel Assignment, Open Label, Treatment	29-Dec-23	Incyte Corporation	Various, United States, Belgium, France, Italy, Spain, United Kingdom
11	NCT03556228	Phase 1	≥18 years	74	Interventional; Sequential Assignment, Open Label, Treatment	Jun-24	VM Oncology, LLC	Various, United States
12	NCT03805594	Phase 1	≥18 years	43	Interventional; Non-Randomized, Open Label, Treatment	30-Apr-24	University of California, San Francisco; Prostate Cancer Foundation; National Cancer Institute (NCI)	United States
13	NCT05077098	Phase 1	18–99 Years	21	Interventional; Single Group, Open Label, Treatment	Sep-24	Mark Stein; Columbia University	United States
14	NCT03792841	Phase 1	≥18 years	212	Interventional; Non-Randomized, Open Label, Treatment	16-May-25	Amgen	Various, Australia, Austria, Belgium, Canada, Japan, Singapore, Taiwan
15	NCT04477512	Phase 1	≥18 years	22	Interventional; Non-Randomized, Open Label, Treatment	31-Aug-25	Washington University School of Medicine; Bristol-Myers Squibb; Exelixis	United States
16	NCT04514484	Phase 1	≥18 years	18	Interventional; Single Group, Open Label, Treatment	2-Nov-25	National Cancer Institute (NCI)	United States
17	NCT05354375	Phase 1	18–75 Years	20	Interventional; Single Group, Open Label, Treatment	30-Nov-26	The Affiliated Hospital of Xuzhou Medical University; Xuzhou Medical University	China
18	NCT04221542	Phase 1	≥18 years	459	Interventional; Non-Randomized, Open Label, Treatment	9-Aug-27	Amgen	Various, United States, Australia, Japan, Korea, Taiwan
19	NCT05010200	Phase 1	≥18 years	27	Interventional; Non-Randomized, Open Label, Prevention	Dec-27	Ashutosh Kumar Tewari; Icahn School of Medicine at Mount Sinai	United States

**Table 4 jcm-12-01446-t004:** Characteristics of Current Clinical Trials at Phase I/II for Prostate Cancer, 2022 (as of 5 December 2022).

No.	NCT Number	Status	Conditions	Interventions	Key Outcome Measures
1	NCT03658447	Active, not recruiting	mCRPC	Pembrolizumab; 177Lu-PSMA	PSA; treatment-emergent adverse events (safety); tolerability; Radiographic PFS; ORR DOR; TTR response; pain; quality of life
2	NCT04071236	Recruiting	CRPC; Metastatic Malignant Neoplasm in the Lymph Nodes; Metastatic Prostate Carcinoma; Stage IV Prostate Cancer AJCC v8	Avelumab; Peposertib; Radium Ra 223 Dichloride	DLTs, PFS; OS; SSE; toxicity and adverse events
3	NCT04382898	Active, not recruiting	Prostate Cancer	BNT112; Cemiplimab	DLTs; adverse events; ORR; PSA levels; change in PSA doubling time
4	NCT03689699	Active, not recruiting	Prostate Cancer; Adenocarcinoma	Nivolumab; Degarelix; BMS-986253	PSA; safety and tolerability; RFS
5	NCT01688492	Active, not recruiting	Prostate Cancer	Ipilimumab	PFS; PSA kinetics; changes in radionuclide bone scan
6	NCT03217747	Active, not recruiting	CRPC; Metastasis in the Bone; Stage IV, IVA, IVB Prostate Cancer AJCC v8;	Avelumab; Ivuxolimab; Radiation Therapy; Utomilumab	Adverse events; CD8 immune biomarkers in tumor and blood; ORR; PFS; DOR; OS
7	NCT02933255	Recruiting	Prostate Cancer	PROSTVAC-V/F; Nivolumab	Safety; immune cell changes; T cells in the tumor; pathologic responses; PSA changes; MRI changes
8	NCT03493945	Recruiting	Metastatic Prostate Cancer; Prostate Cancer; Advanced Solid Tumor; Solid Tumor	M7824; N-803; MVA-BN-Brachyury; FPV-Brachyury; Epacadostat	PFS; safety profile
9	NCT05000294	Recruiting	Prostate Cancer and various others	Atezolizumab; Tivozanib	ORR; PFS; OS; DCR
10	NCT03518606	Active, not recruiting	Prostate Cancer and various others	Durvalumab; Tremelimumab; Metronomic Vinorelbine	MTD and RP2D
11	NCT03543189	Recruiting	Prostate Cancer	Nivolumab; Brachytherapy; External Beam Radiation Therapy; Androgen Deprivation Therapy	Safety; DLTs; RFS; PSA
12	NCT04109729	Recruiting	mCRPC	Nivolumab	Safety; PSA; PFS; bone metabolism markers; SSE
13	NCT05553639	Not yet recruiting	Prostate Cancer Metastatic	HB-302/HB-301 Alternating 2-Vector Therapy	N/A
14	NCT04946370	Recruiting	Prostate Cancer	225Ac-J591; Pembrolizumab; Androgen receptor pathway inhibitor	DLTs; composite response rate; OS; PFS; OS; PSA

Abbreviations: CRPC: castration-resistant prostate cancer; DCR: disease control rate; DLTs: dose-limiting toxicities; DOR: duration of response; mCRPC: metastatic castrate-resistant prostate carcinoma; MTD: maximum tolerated dose; ORR: overall response rate; OS: overall survival; PFS: progression-free survival; PSA: prostate-specific antigen; RFS: Relapse-free survival; RP2D: Phase II recommended dose; SSE: symptomatic skeletal event; TTF: time to treatment failure; TTR: time to treatment.

**Table 5 jcm-12-01446-t005:** Study Design, Funding, Enrollment, and Key Locations of Current Clinical Trials at Phase I/II for Prostate Cancer, 2022 (as of 5 December 2022).

No.	NCT Number	Phases	N	Study Type and Design	Completion Date	Collaborators	Locations
1	NCT03658447	Phase 1, 2	37	Interventional; Single Group, Open Label, Treatment	Dec-22	Peter MacCallum Cancer Centre, Australia	Australia
2	NCT04071236	Phase 1, 2	90	Interventional; Randomized, Open Label, Treatment	31-Jan-23	National Cancer Institute (NCI)	Various, United States
3	NCT04382898	Phase 1, 2	115	Interventional; Randomized, Open Label, Treatment	Jul-23	BioNTech SE	Various, United States, Germany, Hungary, United Kingdom
4	NCT03689699	Phase 1, 2	60	Interventional; Randomized, Open Label, Treatment	Aug-23	Matthew Dallos; Bristol-Myers Squibb; Columbia University	United States
5	NCT01688492	Phase 1, 2	57	Interventional; Single Group, Open Label, Treatment	Sep-23	Memorial Sloan Kettering Cancer Center; Bristol-Myers Squibb; Northwestern University; Oregon Health and Science University	United States
6	NCT03217747	Phase 1, 2	173	Interventional; Non-Randomized, Open Label, Treatment	30-Sep-23	M.D. Anderson Cancer Center; National Cancer Institute (NCI)	United States
7	NCT02933255	Phase 1, 2	29	Interventional; Non-Randomized, Open Label, Treatment	1-Dec-23	National Cancer Institute (NCI); National Institutes of Health Clinical Center (CC)	United States
8	NCT03493945	Phase 1, 2	113	Interventional; Randomized, Open Label, Treatment	31-Dec-23	National Cancer Institute (NCI); National Institutes of Health Clinical Center (CC)	United States
9	NCT05000294	Phase 1, 2	29	Interventional; Sequential Assignment, Open Label, Treatment	Jun-24	University of Florida; Genentech, Inc.; Aveo Oncology Pharmaceuticals	United States
10	NCT03518606	Phase 1, 2	150	Interventional; Non-Randomized, Open Label, Treatment	30-Dec-24	UNICANCER; National Cancer Institute, France; AstraZeneca; Pierre Fabre Laboratories	Various, France
11	NCT03543189	Phase 1, 2	44	Interventional; Single Group, Open Label, Treatment	Dec-24	H. Lee Moffitt Cancer Center and Research Institute; Bristol-Myers Squibb	United States
12	NCT04109729	Phase 1, 2	36	Interventional; Single Group, Open Label, Treatment	30-Apr-25	University of Utah	United States
13	NCT05553639	Phase 1, 2	70	Interventional; Single Group, Open Label, Treatment	Sep-26	Hookipa Biotech GmbH	United States
14	NCT04946370	Phase 1, 2	76	Interventional; Randomized, Open Label, Treatment	Jun-28	Weill Medical College of Cornell University; United States Department of Defense; Merck Sharp & Dohme LLC	Various, United States

**Table 6 jcm-12-01446-t006:** Characteristics of Current Clinical Trials at Phases II and III for Prostate Cancer, 2022 (as of 5 December 2022).

No.	NCT Number	Status	Conditions	Interventions	Key Outcome Measures
1	NCT03207867	Active, not recruiting	mCRPC	NIR178; PDR001	ORR; DCR; DOR; OS; PFS; safety and tolerability; pharmacokinetics
2	NCT03866382	Recruiting	Metastatic Prostate Small-Cell Neuroendocrine Carcinoma; Stage IV, IVA, IVB Prostate Cancer AJCC v8	Cabozantinib S-malate; Ipilimumab; Nivolumab	ORR; DOR; PFS; OS; CBR; adverse events
3	NCT02768363	Active, not recruiting	Prostate Cancer	Aglatimagene besadenovec; valacyclovir	PFS; PSA; time to radical treatment; adverse events
4	NCT04104893	Recruiting	mCRPC	Pembrolizumab	PSA; ORR; time to progression of disease; OS; adverse events (safety and tolerability)
5	NCT03651271	Active, not recruiting	Advanced Metastatic Cancer; Advanced Prostate Cancer	Nivolumab Monotherapy; Nivolumab + Ipilimumab	CBR; CD8 cells in biopsies; safety; ORR
6	NCT03570619	Active, not recruiting	mCRPC; Metastatic Cancer; Solid Tumor	Nivolumab; Ipilimumab	Patient response with CDK12 loss of function to treatment; PFS; TTP; OS; PSA
7	NCT02703623	Active, not recruiting	CRPC; Metastatic PC; PSA Progression; Stage IV Prostate Adenocarcinoma AJCC v7	Abiraterone Acetate; Apalutamide; Cabazitaxel; Carboplatin; Ipilimumab; Prednisone	OS; adverse events; androgen receptor response markers signature; TTF
8	NCT04009967	Recruiting	Prostate Cancer	Pembrolizumab	Tumor response rate; Immune parameters; PSA; correlation of dMMR/MSI-H with pembrolizumab response
9	NCT03338790	Active, not recruiting	Prostate Cancer	Nivolumab; docetaxel; enzalutamide; rucaparib; prednisone	ORR; PSA; PFS; time to response; DOR; adverse events; deaths; laboratory abnormalities
10	NCT03821246	Recruiting	Prostate Adenocarcinoma; Localized Prostate Cancer	Atezolizumab; Tocilizumab; Etrumadenant	Positive response; adverse events; pCR; MRD; PSA response
11	NCT05177770	Recruiting	mCRPC	SRF617; Etrumadenant; Zimberelimab	PSA; adverse events; ORR; DOR; DCR; pharmacokinetics; SSEs
12	NCT03315871	Recruiting	Prostate Cancer	PROSTVAC-V; PROSTVAC-F; MSB0011359C (M7824); CV301	PSA; adverse events
13	NCT02020070	Active, not recruiting	Metastatic Castration-Sensitive Prostate Cancer	Degarelix; Ipilimumab; Radical Prostatectomy	PSA; PFS; OS; toxicity
14	NCT03385655	Recruiting	Prostate Cancer	Adavosertib; Savolitinib; Darolutamide; CFI-400945; Ipatasertib; Durvalumab and Tremelimumab; Carboplatin	PSA decline of 50%; PSA progression; objective response; adverse events; PFS; OS
15	NCT03764540	Recruiting	Metastatic Prostate Cancer	Cabazitaxel plus prednisone; Docetaxel plus prednisone	PSA response rate; PFS; OS; TTP; tumor response; DOR; pain response
16	NCT05502315	Not yet recruiting	CRPC; Metastatic Cancer	Cabozantinib; Nivolumab	PFS; ORR; OS; CTC; adverse events; SSEs
17	NCT03795207	Recruiting	Node/Bone Metastases; Prostate Cancer	SBRT + Durvalumab	PFS; ADT free survival; OS; acute toxicity; time to castration resistance
18	NCT05361798	Recruiting	Prostate Cancer	M9241; SBRT	Safety; T cell clonality (immunologic activity); peripheral immune response
19	NCT04751929	Recruiting	Prostate Cancer; mCRPC	Abemaciclib; Atezolizumab	PFS; ORR; DLTs; adverse events; CBR; DOR; DOT; TTP; OS
20	NCT04336943	Recruiting	Biochemically Recurrent Prostate Carcinoma; Prostate Adenocarcinoma	Durvalumab; Olaparib	PSA; adverse events; quality of life
21	NCT03333616	Recruiting	Non-adenocarcinoma Prostate Cancer, and various others	Ipilimumab; Nivolumab	ORR; DOR; OS; safety and tolerability; adverse events
22	NCT04717154	Recruiting	Prostatic Neoplasms, Castration-Resistant	Ipilimumab; Nivolumab	DCR; adverse effects; ORR; PFS
23	NCT04126070	Recruiting	Hormone-Sensitive Prostate Cancer; Prostate Adenocarcinoma; Metastasis Prostate Adenocarcinoma	ADT; Nivolumab; Docetaxel	PSA; ORR; OR; time to castration resistance/clinical progression/serologic progression; severe adverse events
24	NCT04592237	Recruiting	Aggressive PC; CRPC; Metastatic Prostate Carcinoma; Metastatic Prostate Neuroendocrine Carcinoma; Metastatic Prostate Small-Cell Carcinoma; Stage IV Prostate Cancer AJCC v8	Cabazitaxel; Carboplatin; Cetrelimab; Niraparib	PFS; OS; RR; adverse events
25	NCT04090528	Recruiting	CRPC; Metastatic Cancer; Prostate Cancer	pTVG-HP; pTVG-AR; Pembrolizumab	PFS; ORR; PSA; RR; OS; antigen-specific Th1 immune response; safety and tolerability
26	NCT04926181	Recruiting	Small Cell Neuroendocrine Carcinoma; Prostate Cancer; Small-Cell Carcinoma	Apalutamide; Cetrelimab	Composite RR; adverse events; median PFS; PSA; OS; OSS; DOR
27	NCT05445882	Not yet recruiting	CRPC	Bintrafusp alfa; N-803; BN-Brachyury	Clinical efficacy; DOR; safety
28	NCT05168618	Recruiting	CRPC; Metastatic Prostate Adenocarcinoma; Stage IV, IVA, IVB Prostate Cancer AJCC v8	Atezolizumab; Cabozantinib S-malate	DCR; PSA; PFS; OS; adverse events
29	NCT05568550	Not yet recruiting	Prostate Cancer	Pembrolizumab; Olaparib; ADT; Radiation Therapy	Clinical RR; biochemical/metastasis-free survival; molecular alterations in homologous recombination repair genes
30	NCT03879122	Active, not recruiting	Metastatic Hormone-sensitive Prostate Cancer	Ipilimumab 5 MG/ML; Nivolumab 10 MG/ML; Docetaxel; ADT	OS; PSA; PFS; time to CRPC; PFS; SSEs; toxicity; quality of life
31	NCT01436968	Active, not recruiting	Prostate Cancer	Aglatimagene besadenovec + valacyclovir; Placebo + valacyclovir	DFS; OS; PSA; safety; quality of life
32	NCT03686683	Active, not recruiting	Adenocarcinoma, Prostate	Sipuleucel-T	Efficacy in reducing histopathologic reclassification to a higher Gleason grade
33	NCT05544227	Recruiting	mCRPC	SV-102	Anti-tumor activity; adverse events; treatment discontinuation
34	NCT05544240	Recruiting	mCRPC	SV-101	Anti-tumor activity; adverse events; treatment discontinuation
35	NCT02971358	Recruiting	Locally Advanced and Metastatic Prostate Cancer	Radical prostatectomy	Perioperative complications; time to start ADT

Abbreviations: ADT: androgen deprivation therapy; CBR: clinical benefit rate; CRPC: castration-resistant prostate cancer; CTC: circulating tumor cells; DCR: disease control rate; DFS: disease-free survival; dMMR: deficient mismatched repair; DOR: duration of response; DOT: duration of therapy; mCRPC: metastatic castrate-resistant prostate carcinoma; MRD: minimal residual disease; MSI-H: microsatellite instability-high; ORR: overall response rate; OS: overall survival; pCR: pathological complete responses; PFS: progression-free survival; PSA: prostate-specific antigen; RR: response rate; SBRT: stereotactic body radiation therapy; SSE: symptomatic skeletal event; TTF: time to treatment failure; TTP: time to progression.

**Table 7 jcm-12-01446-t007:** Study Design, Funding, Enrollment, and Key Locations of Current Clinical Trials at Phase II and III for Prostate Cancer, 2022 (as of 5 December 2022).

No.	NCT Number	Phases	N	Study Type and Design	Completion Date	Collaborators	Locations
1	NCT03207867	Phase 2	317	Interventional; Non-Randomized, Open Label, Treatment	21-Dec-22	Novartis Pharmaceuticals; Novartis	Various, United States, Argentina, Australia, Belgium, Czechia, France, Germany, Italy, Japan, Netherlands, Singapore, Spain, Switzerland, Taiwan
2	NCT03866382	Phase 2	224	Interventional; Single Group, Open Label, Treatment	28-Feb-23	National Cancer Institute (NCI)	United States
3	NCT02768363	Phase 2	187	Interventional; Randomized; Quadruple Masking, Treatment	Mar-23	Candel Therapeutics, Inc.	Various, United States, Mexico
4	NCT04104893	Phase 2	30	Interventional; Single Group, Open Label, Treatment	31-Mar-23	VA Office of Research and Development; Merck Sharp & Dohme LLC	Various, United States
5	NCT03651271	Phase 2	220	Interventional; Non-Randomized, Open Label, Treatment	May-23	Parker Institute for Cancer Immunotherapy; Bristol-Myers Squibb; Cancer Research Institute, New York City	Various, United States
6	NCT03570619	Phase 2	65	Interventional; Non-Randomized, Open Label, Treatment	May-23	University of Michigan Rogel Cancer Center; Memorial Sloan Kettering Cancer Center; University of California, San Francisco	Various, United States
7	NCT02703623	Phase 2	196	Interventional; Randomized, Open Label, Treatment	18-May-23	M.D. Anderson Cancer Center; National Cancer Institute (NCI)	United States
8	NCT04009967	Phase 2	30	Interventional; Single Group, Open Label, Treatment	30-May-23	CHU de Quebec-Universite Laval; Merck Sharp & Dohme LLC	Canada
9	NCT03338790	Phase 2	292	Interventional; Non-Randomized, Open Label, Treatment	15-Jul-23	Bristol-Myers Squibb; Clovis Oncology, Inc.; Astellas Pharma Inc	Various, United States, Australia, Brazil, Canada, Chile, France, Germany, Mexico, Spain
10	NCT03821246	Phase 2	68	Interventional; Non-Randomized, Open Label, Treatment	31-Oct-23	Lawrence Fong; Genentech, Inc.; University of California, San Francisco	United States
11	NCT05177770	Phase 2	40	Interventional; Single Group, Open Label, Treatment	Nov-23	Surface Oncology; Arcus Biosciences, Inc.	Various, United States
12	NCT03315871	Phase 2	40	Interventional; Non-Randomized, Open Label, Treatment	1-Dec-23	National Cancer Institute (NCI); National Institutes of Health Clinical Center (CC)	United States
13	NCT02020070	Phase 2	16	Interventional; Non-Randomized, Open Label, Treatment	Dec-23	Memorial Sloan Kettering Cancer Center; Ferring Pharmaceuticals	United States
14	NCT03385655	Phase 2	500	Interventional; Non-Randomized, Open Label, Treatment	31-Jul-24	Canadian Cancer Trials Group; Canadian Cancer Clinical Trials Network; BC Cancer Foundation	Various, Canada
15	NCT03764540	Phase 2	214	Interventional; Randomized, Open Label, Treatment	30-Sep-24	Centre hospitalier de l’UniversitÃ de Montreal (CHUM); Genzyme, a Sanofi Company	Canada
16	NCT05502315	Phase 2	50	Interventional; Single Group, Open Label, Treatment	12-Oct-24	Rana McKay, MD; Exelixis; Bristol-Myers Squibb; Hoosier Cancer Research Network	N/A
17	NCT03795207	Phase 2	96	Interventional; Randomized, Open Label, Treatment	21-Nov-24	Institut Cancerologie de l’Ouest; AstraZeneca	Various, France
18	NCT05361798	Phase 2	65	Interventional; Randomized, Open Label, Treatment	1-Dec-24	National Cancer Institute (NCI); National Institutes of Health Clinical Center (CC)	United States
19	NCT04751929	Phase 2	75	Interventional; Randomized, Open Label, Treatment	20-Dec-24	Dana-Farber Cancer Institute; Eli Lilly and Company; Genentech, Inc.	United States
20	NCT04336943	Phase 2	30	Interventional; Single Group, Open Label, Treatment	30-Apr-25	University of Washington; AstraZeneca	United States
21	NCT03333616	Phase 2	100	Interventional; Single Group, Open Label, Treatment	31-May-25	Dana-Farber Cancer Institute; Bristol-Myers Squibb	Various, United States
22	NCT04717154	Phase 2	75	Interventional; Single Group, Open Label, Treatment	30-Jun-25	Radboud University Medical Center; Bristol-Myers Squibb	The Netherlands
23	NCT04126070	Phase 2	60	Interventional; Non-Randomized, Open Label, Treatment	30-Jun-25	Xiao X. Wei; Bristol-Myers Squibb; Dana-Farber Cancer Institute	United States
24	NCT04592237	Phase 2	120	Interventional; Randomized, Open Label, Treatment	31-Dec-25	M.D. Anderson Cancer Center; Janssen Pharmaceuticals	United States
25	NCT04090528	Phase 2	60	Interventional; Randomized, Open Label, Treatment	Dec-25	University of Wisconsin, Madison; Merck Sharp & Dohme LLC; Madison Vaccines, Inc; Prostate Cancer Foundation	United States
26	NCT04926181	Phase 2	24	Interventional; Single Group, Open Label, Treatment	31-May-26	Rahul Aggarwal; Janssen Scientific Affairs, LLC; University of California, San Francisco	United States
27	NCT05445882	Phase 2	28	Interventional; Non-Randomized, Open Label, Treatment	1-Aug-26	National Cancer Institute (NCI); National Institutes of Health Clinical Center (CC)	United States
28	NCT05168618	Phase 2	33	Interventional; Single Group, Open Label, Treatment	Jan-27	University of Utah; National Cancer Institute (NCI)	United States
29	NCT05568550	Phase 2	64	Interventional; Randomized, Open Label, Treatment	2-Jan-28	Zin W Myint; Merck Sharp & Dohme LLC; University of Kentucky	United States
30	NCT03879122	Phase 2, 3	135	Interventional; Randomized, Open Label, Treatment	31-Dec-24	Spanish Oncology Genito-Urinary Group; Syntax for Science, S.L; Bristol-Myers Squibb	Various, Spain
31	NCT01436968	Phase 3	711	Interventional; Randomized; Quadruple Masking, Treatment	Jun-2023	Candel Therapeutics, Inc.	Various, United States, Puerto Rico
32	NCT03686683	Phase 3	450	Interventional; Randomized, Open Label, Treatment	May-23	Dendreon; PRA Health Sciences	Various, United States
33	NCT05544227	Not Applicable	20	Interventional; Single Group, Open Label, Treatment	31-Dec-23	Williams Cancer Foundation; Syncromune, Inc.	Mexico
34	NCT05544240	Not Applicable	20	Interventional; Single Group, Open Label, Treatment	31-Dec-23	Williams Cancer Foundation; Syncromune, Inc.	Mexico
35	NCT02971358	Not Applicable	200	Interventional; Single Group, Open Label, Treatment	Dec-27	Medical University of Vienna	Austria

## Data Availability

Not applicable.

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
