# Peer review of "Immunotherapy for Prostate Cancer: A Current Systematic Review and Patient Centric Perspectives"

_jcm, 2023, doi:10.3390/jcm12041446_

Round 1
Reviewer 1 Report
In this manuscript, the Authors aimed to review systematically the available literature to first collate and synthesize findings of all completed Phase III clinical trials administering immunotherapy to patients with prostate cancer and secondly formulate a current clinical trial index (2022) of all Phase I-III clinical trial records that are ongoing in the field.
Although the Authors reported a large amount of information, the methodology of the systematic review has some limitations, and the results are not displayed in a well-organized way.
Here I report my suggestions:
Major Revisions:
-“Introduction” section: I suggest reducing the length of this section. I think the role of an “introduction” section in a systematic review is to explain briefly to the reader the rationale that led to the PICO question of the review, and why the Authors decided to revise the literature. In this direction, a first step may be represented by removing the text from line 49 to 80 because it is just an overview of the prostate cancer treatment, not related to the field of immunotherapy in prostate cancer. If the Authors consider this information important, they could summarize them in the “Discussion” section;
-“Introduction” section: There are a lot of reviews on immunotherapy in prostate cancer available in the literature. I suggest clarifying in more details why the Authors decided to focus only on phase III trials and on ongoing trials.
-“Methods” section: A systematic review performed according to PRISMA guidelines requires a “Study risk of bias assessment”. I suggest reporting in the Methods how the Authors performed the risk of bias assessment. Consequently, I suggest reporting the results of the assessment in the “Results” section, providing in the Supplementary Material the evidence of this evaluation.
-Line 107-108: I suggest clarifying why the Authors searched single journal databases, since they evaluated PubMed.
-“Results”: Why the Authors inserted among the phase III trial the study led by Sartor et al. (2021, NEJM)? 177Lu-PSMA-617 delivers beta-particle radiation selectively to PSMA-positive cells and the surrounding microenvironment and thus it is a radioligand and not an immunotherapeutic strategy;
-Tables: I suggest reducing the number of the words in the Tables without losing important information. It could be useful to use abbreviations. In addition, I suggest removing the entire title (which can be retrieved with the related reference) and organize better the information available in the other columns. Furthermore, I suggest putting together Table 2 and Table 3; Table 4 with Table 5; Table 6 with Table 7. Indeed, it is hard for the reader to retrieve the information when there are two different tables on the same clinical trials.
Minor Revisions:
-“Literature search”paragraph: I suggest uploading a PRISMA Checklist as a Supplementary Material where the Authors confirm that all the PRISMA requirements were followed (https://prisma-statement.org/prismastatement/Checklist.aspx)
-“Literature search”paragraph: I suggest clarifying if a protocol was designed a priori.
-“Literature search”paragraph: Was a specific “search strategy” used? If yes, I suggest reporting in the manuscript or as a Supplementary Material
-Line 127: I suggest reporting the name of the software;
-Lines 142 to 150: I suggest removing the number in brackets to list the variable extracted.
- Extensive editing of English language and style required
Author Response
To the Reviewer: I thank you for your time and attention given to review our paper. My responses to all your comments are appended below.
Kind Regards,
Zouina S.
Reviewer 1 Comments and Author Responses:
In this manuscript, the Authors aimed to review systematically the available literature to first collate and synthesize findings of all completed Phase III clinical trials administering immunotherapy to patients with prostate cancer and secondly formulate a current clinical trial index (2022) of all Phase I-III clinical trial records that are ongoing in the field.
Although the Authors reported a large amount of information, the methodology of the systematic review has some limitations, and the results are not displayed in a well-organized way.
Here I report my suggestions:
Major Revisions:
Comment 1:
-“Introduction” section: I suggest reducing the length of this section. I think the role of an “introduction” section in a systematic review is to explain briefly to the reader the rationale that led to the PICO question of the review, and why the Authors decided to revise the literature. In this direction, a first step may be represented by removing the text from line 49 to 80 because it is just an overview of the prostate cancer treatment, not related to the field of immunotherapy in prostate cancer. If the Authors consider this information important, they could summarize them in the “Discussion” section;
Author Response to Comment 1:
Thank you for your suggestion. A rationale has been added to the introduction which has been generated in line with your comment. I invite you to review the amendments. The length of the introduction has been reduced as well with reordering of text.
Comment 2:
-“Introduction” section: There are a lot of reviews on immunotherapy in prostate cancer available in the literature. I suggest clarifying in more detail why the Authors decided to focus only on phase III trials and on ongoing trials.
Author Response to Comment 2:
Thank you for your suggestion. A new paragraph has been added to the introduction highlighting the reasoning behind our decision to only focus on Phase III trials. I invite you to review the changes.
Comment 3:
-“Methods” section: A systematic review performed according to PRISMA guidelines requires a “Study risk of bias assessment”. I suggest reporting in the Methods how the Authors performed the risk of bias assessment. Consequently, I suggest reporting the results of the assessment in the “Results” section, providing in the Supplementary Material the evidence of this evaluation.
Author Response to Comment 3:
Risk of bias information has been added in the methodology. Moreover, I fully agree with your comment. A new section has been made in the results section (3.3) where a Figure 2 has been added as well. I invite you to review the additions.
Comment 4:
-Line 107-108: I suggest clarifying why the Authors searched single journal databases, since they evaluated PubMed.
Author Response to Comment 4:
Please review the methodology for the reasoning. It is as follows: “An additional search was conducted in Elsevier, the BMJ, JAMA, NEJM, and the Lan-cet to locate relevant literature; this methodology is referred to as handsearching, and is utilized to identify any additional randomized controlled trials administering im-munotherapy to patients with prostate cancer.”
Comment 5:
-“Results”: Why the Authors inserted among the phase III trial the study led by Sartor et al. (2021, NEJM)? 177Lu-PSMA-617 delivers beta-particle radiation selectively to PSMA-positive cells and the surrounding microenvironment and thus it is a radioligand and not an immunotherapeutic strategy;
Author Response to Comment 5:
Thank you for noting a major issue in the included study. It has been removed as it does not constitute immunotherapy intervention. Subsequently, the entire paper has been updated to reflect the removal. The PRISMA flow chart along with all associated sections have been updated too.
Comment 6:
-Tables: I suggest reducing the number of the words in the Tables without losing important information. It could be useful to use abbreviations. In addition, I suggest removing the entire title (which can be retrieved with the related reference) and organize better the information available in the other columns. Furthermore, I suggest putting together Table 2 and Table 3; Table 4 with Table 5; Table 6 with Table 7. Indeed, it is hard for the reader to retrieve the information when there are two different tables on the same clinical trials.
Author Response to Comment 6:
Thank you for your comment. I have removed the titles from the tables. However, the tables as you have suggested cannot be joined together. Their width will make them unreadable. Therefore, the tables remain divided. The number of words have already been minimized and further reduction will lead to missing clinical information.
Minor Revisions:
Comment 7:
-“Literature search” paragraph: I suggest uploading a PRISMA Checklist as a Supplementary Material where the Authors confirm that all the PRISMA requirements were followed (https://prisma-statement.org/prismastatement/Checklist.aspx)
Author Response to Comment 7:
Thank you for your essential suggestion. The PRISMA 2020 Checklist has been attached under Supplementary Materials. I invite you to review the update.
Comment 8:
-“Literature search” paragraph: I suggest clarifying if a protocol was designed a priori.
Author Response to Comment 8:
The protocol information has been updated in the manuscript. The protocol of this systematic review was registered with Open Science Frame-work (OSF): osf.io/4vs7w.
Comment 9:
-“Literature search” paragraph: Was a specific “search strategy” used? If yes, I suggest reporting in the manuscript or as a Supplementary Material
Author Response to Comment 9:
Thank you for your essential suggestion. The ‘Search String’ has been attached under Supplementary Materials. I invite you to review the update.
Comment 10:
-Line 127: I suggest reporting the name of the software;
Author Response to Comment 10:
Thank you for your comment. The software has been listed. I invite you to review the update.
Comment 11:
-Lines 142 to 150: I suggest removing the number in brackets to list the variable extracted.
Author Response to Comment 11:
Thank you for your comment. The brackets have been removed. I invite you to review the update.
Comment 12:
- Extensive editing of English language and style required
Author Response to Comment 12:
Thank you for your comment. The paper has been proofread by our English Native Speakers/Writers. I believe there are no central issues with the English language and style. Should any minor issues arise, they will be updated by the editorial team during proofing. I thank you for your due diligence on this matter.
Reviewer 2 Report
In this study authors evaluated the differences in impact on maternal renal function between singleton and twin pregnancies in the second half of pregnancy. Authors found that twin pregnancy, nulliparity and preeclampsia were significant risk factors for maternal renal dysfunction. Thus, authors suggested that renal function should be carefully monitored in twin pregnancies.
The manuscript showed important results that should be evaluated in clinical practice. The manuscript is interesting, clear and generally well written.
Author Response
Reviewer 2 comment and author response:
In this study authors evaluated the differences in impact on maternal renal function between singleton and twin pregnancies in the second half of pregnancy. Authors found that twin pregnancy, nulliparity and preeclampsia were significant risk factors for maternal renal dysfunction. Thus, authors suggested that renal function should be carefully monitored in twin pregnancies.
The manuscript showed important results that should be evaluated in clinical practice. The manuscript is interesting, clear and generally well written.
Author Response: Thank you for your comments.
Round 2
Reviewer 1 Report
I congratulate the Authors for the great work done to improve the manuscript.
Here I report my suggestions/comments for this second review report:
-Introduction: The Authors improved the Introduction section as suggested in the previous review report. However, this paragraph is now composed of three subparagraphs: the first one without a title, the second one named "Rationale," and the third one called "Aims and objective". I suggest organizing these three subparagraphs in just two different subparagraphs named "Rationale" and "Objectives", following the PRISMA guidelines. This type of organization will contribute to further reducing the information reported in this first part of the paper, focusing only on the most valuable data in introducing the reader to the design of the manuscript.
Introduction: I suggest changing Lines 77-78. Indeed, the only immunotherapeutic treatment approved for PCa is Sipuleucel-T. The other treatments available are related to the agnostic approval of pembrolizumab and dostarlimab for solid metastatic tumors harboring conditions of genomic hypermutability (dMMR/MSI-H or TMB-h). In this direction, I suggest including this reference for the role of agnostic treatments in PCa (PMID: 35955671).
-Lines 122-123: I suggest highlighting in the limitations of the study that "Google Translate" was used to translate non-English manuscripts. Although this tool is widely used, it cannot substitute for an interpreter specialized in medical research, which should be the gold standard for publications of medical manuscripts.
Line 155-156: I suggest changing the link provided. In fact, it requires a login to read the protocol. Therefore, I suggest adding a link that gives the reader access to the protocol without asking for the login in OSF.
-Tables: Although the Authors improved the Tables, I suggest a further revision to reduce the number of words included in the cells of the columns. I suggest using abbreviations and further summarizing the content of the columns, especially "Design”, “Inclusion criteria”, “Intervention" for Table 1; "Outcome Measures", "Study design", "Locations" (for example, a potential strategy would be to summarize the locations in Asia vs USA vs Europe vs rest of the world like in demographics of clinical trials) for other Tables;
-Discussion: In the previous review report, I suggested the Authors remove Lines "308-339" from the Introduction, and I wrote "If the Authors consider this information important, they could summarize them in the “Discussion” section". The Authors did not summarize these Lines and just moved the entire paragraph to the Discussion section. This part represents an overview of the treatment of prostate cancer without any link to the role of immunotherapy in this disease. As a result, I suggest removing this entire part from the manuscript.
-Discussion: I suggest adding a subparagraph in the Discussion named "Limitations" where the Authors report the limitations of this study. Accordingly, I suggest modifying the PRISMA checklist.
-Discussion: I suggest including these two publications in the review to enrich the future perspective of immunotherapy of prostate cancer (PMID: 35181473 and PMID: 35158771)
Author Response
Reviewer Comments and Author Responses:
Comment 1: Introduction: The Authors improved the Introduction section as suggested in the previous review report. However, this paragraph is now composed of three subparagraphs: the first one without a title, the second one named "Rationale," and the third one called "Aims and objective". I suggest organizing these three subparagraphs in just two different subparagraphs named "Rationale" and "Objectives", following the PRISMA guidelines. This type of organization will contribute to further reducing the information reported in this first part of the paper, focusing only on the most valuable data in introducing the reader to the design of the manuscript.
Response to Comment 1: Thank you for your comment. However, on deliberation with the entire author team, it would be incorrect to completely remove the first paragraph before the rationale section. It does not make sense for us to remove this valuable piece of information when writing on such an important topic.
Comment 2: Introduction: I suggest changing Lines 77-78. Indeed, the only immunotherapeutic treatment approved for PCa is Sipuleucel-T. The other treatments available are related to the agnostic approval of pembrolizumab and dostarlimab for solid metastatic tumors harboring conditions of genomic hypermutability (dMMR/MSI-H or TMB-h). In this direction, I suggest including this reference for the role of agnostic treatments in PCa (PMID: 35955671).
Response to Comment 2: Thank you for your comment. The reference has been added along with new supporting information. Please review the changes highlighted in green.
Comment 3: Lines 122-123: I suggest highlighting in the limitations of the study that "Google Translate" was used to translate non-English manuscripts. Although this tool is widely used, it cannot substitute for an interpreter specialized in medical research, which should be the gold standard for publications of medical manuscripts.
Response to Comment 3: Thank you for your comment. The limitation has been added.
Comment 4: Line 155-156: I suggest changing the link provided. In fact, it requires a login to read the protocol. Therefore, I suggest adding a link that gives the reader access to the protocol without asking for the login in OSF.
Response to Comment 4: The protocol has been ‘made public’ in the OSF registry for viewing now. It should not be a problem here onwards. Here is the public DOI as well: Identifier: DOI 10.17605/OSF.IO/4VS7W.
Comment 5: Tables: Although the Authors improved the Tables, I suggest a further revision to reduce the number of words included in the cells of the columns. I suggest using abbreviations and further summarizing the content of the columns, especially "Design”, “Inclusion criteria”, “Intervention" for Table 1; "Outcome Measures", "Study design", "Locations" (for example, a potential strategy would be to summarize the locations in Asia vs USA vs Europe vs rest of the world like in demographics of clinical trials) for other Tables;
Response to Comment 5: Table 1 has been updated with as much reduction of words as possible without losing key information. The three columns of “design, inclusion criteria and intervention” have been thoroughly updated. Please note that removing more information would be incorrect on our part. Moreover, all the other tables have been summarized. I have added abbreviations for all the shortened words. Please note that I have retained the countries since the idea is to have them listed as such for the readers to locate them. Thank you for your comment.
Comment 6: Discussion: In the previous review report, I suggested the Authors remove Lines "308-339" from the Introduction, and I wrote "If the Authors consider this information important, they could summarize them in the “Discussion” section". The Authors did not summarize these Lines and just moved the entire paragraph to the Discussion section. This part represents an overview of the treatment of prostate cancer without any link to the role of immunotherapy in this disease. As a result, I suggest removing this entire part from the manuscript.
Response to Comment 6: Thank you for your comment. The section has been removed now.
Comment 7: Discussion: I suggest adding a subparagraph in the Discussion named "Limitations" where the Authors report the limitations of this study. Accordingly, I suggest modifying the PRISMA checklist.
Response to Comment 7: Thank you for your comment. A limitations section has been added.
Comment 8: Discussion: I suggest including these two publications in the review to enrich the future perspective of immunotherapy of prostate cancer (PMID: 35181473 and PMID: 35158771)
Response to Comment 8: Thank you for your comment. Both citations have been added in a new subsection: Future Directions.
Note to Reviewer: Thank you for your time and attention in enriching our paper. I have taken almost all your comments and have gone ahead to make the changes. Very few aspects have been retained since we would like to retain some key objectives of the paper. I trust you will receive this revision well.